# WCLD: Curated Large Dataset of Criminal Cases from Wisconsin Circuit Courts

**Elliott Ash**[1*]   **Naman Goel**[2*]   **Nianyun Li**[1*]   **Claudia Marangon**[1*]   **Peiyao Sun**[1*]

[1]Center for Law & Economics, ETH Zurich
[2]Department of Computer Science, University of Oxford
{elliott.ash,claudia.marangon}@gess.ethz.ch, naman.goel@cs.ox.ac.uk

## Abstract

Machine learning based decision-support tools in criminal justice systems are subjects of intense discussions and academic research. There are important open questions about the utility and fairness of such tools. Academic researchers often rely on a few small datasets that are not sufficient to empirically study various real-world aspects of these questions. In this paper, we contribute WCLD, a curated large dataset of 1.5 million criminal cases from circuit courts in the U.S. state of Wisconsin. We used reliable public data from 1970 to 2020 to curate attributes like prior criminal counts and recidivism outcomes. The dataset contains large number of samples from five racial groups, in addition to information like sex and age (at judgment and first offense). Other attributes in this dataset include neighborhood characteristics obtained from census data, detailed types of offense, charge severity, case decisions, sentence lengths, year of filing etc. We also provide pseudo-identifiers for judge, county and zipcode. The dataset will not only enable researchers to more rigorously study algorithmic fairness in the context of criminal justice, but also relate algorithmic challenges with various systemic issues. We also discuss in detail the process of constructing the dataset and provide a datasheet. The WCLD dataset is available at `https://clezdata.github.io/wcld/`.

## 1   Introduction

Machine learning algorithms are increasingly being used or proposed to be used in various sensitive use-cases, including criminal justice. In such scenarios, a major concern is that the algorithms can perpetuate and amplify different forms of social biases. There is plenty of empirical evidence supporting this concern in the literature (1). For example, the seminal work by ProPublica (2) found COMPAS, a recidivism risk estimation tool, to be unfair against black defendants: making different errors for different demographic groups and at different rates. While any component in the machine learning pipeline can introduce bias and there are also fundamental limitations of algorithms (3; 4), it is also widely acknowledged that training and evaluation datasets are major contributors (5; 6) to unfairness in algorithms. Therefore, it is important for academic researchers to have access to large datasets to study the problem rigorously. Further, the datasets should also contain rich information to allow understanding the interaction of data and algorithmic issues with various systemic issues. A standard research dataset used in this context is the COMPAS dataset (7). While this dataset has been extremely useful so far in bringing attention to some of the issues with machine learning algorithms, it is often limiting in improving our understanding of the algorithms' behavior because of its small size and other limitations (discussed in Section 2). Further, researchers often use synthetic datasets and simulations (8; 9; 10; 11; 12) to complement their analysis. Availability of large scale real-world datasets will bring more grounding to this complementary analysis approach (13; 9).

---

[*]Authors are ordered alphabetically.

In this paper, we contribute a large scale curated dataset of criminal cases from circuit courts in the U.S. state of Wisconsin. The dataset has been constructed from publicly available raw case records, to make that information easy to use in machine learning research. The dataset has a defined set of attributes and prediction variable for research. The attributes include type of offense, sex, race, prior criminal count (for each offense type), age (at judgment and at first offense) and neighbourhood characteristics like population density, proportion who attended college, proportion eligible for food stamp, African American population share, Hispanic population share, proportion of male, proportions who live in rural and urban area and median household income. The prediction variable for research is the observed recidivism within a follow-up period of 2 years, measured in different ways. We also provide more information for each row in the dataset that can be used for further analysis. This information includes charge severity, case decisions, year of filing, sentence length, charge severity for previous cases, sentence lengths for previous cases, pseudo-identifiers for judges, county and zipcode. Mappings for county and zipcode can be made available to the researchers on request. Information that we have excluded from the dataset are name of the defendant, their address, date of birth, dates of case events etc. There are about 1.5 millions rows in the dataset. We will also strive to add more features to this dataset in the future.

We hope that the academic research community will find this dataset useful not only for benchmarking machine learning and fairness algorithms, but also for critically understanding the strengths and weaknesses of various algorithms and how these issues interact with biases due to geography, time, judges, sentencing etc.

In the rest of the paper, we describe the process of constructing the dataset from raw case records accessed through the Wisconsin Circuit Courts Access API. We discuss various choices made while curating the dataset and their relevance in using this dataset for research. We provide summary statistics of various features in the dataset. For reference, we also report the performance of two basic machine learning classifiers (logistic regression and XGBoost) on the dataset. Appendix F contains datasheet (14) for this dataset.

## 2   Related Work

The COMPAS dataset (7) is one of the most frequently used dataset in machine learning fairness literature. It has around 7000 observations from a single court (Broward County, Florida) over two years (2013 and 2014). It has very few samples for racial groups other than Caucasian and African-American. The dataset is limited to the set of defendants assessed with COMPAS at the pre-trial stage, and it does not include information on judges or other officials. The validity of this dataset as a reliable fairness benchmark is often disputed (15).

(16) contribute datasheets for 15 publicly available datasets in criminal justice from the US: such as survey data from victims, policing data, court records, data about enforcement agencies and officers, data about extremists etc. The datsets listed Section 4.4 of their paper are the most relevant in the context of our contribution. However, those are mostly raw case records, not datasets with curated attributes and recidivism outcomes. (13) survey a few other datasets in criminal justice domain that have similar limitations. (17) contribute a dataset that contains information on the personnel, activities, use of force, and complaints in the Chicago Police Department (CPD).

In the field of machine learning fairness more broadly, there are several notable works that introduce new larger datasets or make larger datasets more usable. For example, (18) construct five datasets related to income, employment, health coverage, commute time, and housing. They observe that many results vary with the data highlighting the importance of more data-centric empirical research. (15) release a toolkit to make the Home Mortgage Disclosure Act datasets more easily usable. They also conclude that the variance in small datasets like COMPAS can muddle the reliability of conclusions about algorithmic fairness interventions. We refer the readers to (19) for a detailed survey on datasets used in algorithmic fairness research.

In a recent paper (13), we reported results of a study that was made possible on subsets of the WCLD dataset. In that study, we used simulations to create several semi-synthetic datasets from the WCLD dataset to understand the impact of a few data-centric factors on algorithmic fairness. However, this is only one example and we believe that researchers will use this dataset for understanding various other issues as well. Further, the study in (13) used only some of the information that we are releasing with this dataset. For example, the additional information on judge, county, detailed offense

types (wcisclass), charge severity, probation, detention, sentencing etc can be used for understanding various systemic biases and how they interact with algorithmic fairness interventions. We have now also curated a variable on violent recidivism. With this submission, we also provide a detailed datasheet for the dataset, lists its various limitations and describe in detail the origin and the process of curating the dataset. This datasheet and all the other details will be useful for researchers while considering conducting further research with this dataset, and even to inform decisions about further possible improvements in the dataset.

# 3 Dataset Construction

## 3.1 Wisconsin Circuit Courts Access (WCCA)

WCCA was created in April 1999, and it contains public case records and docket information from the 72 county courts of the U.S. state of Wisconsin. The WCCA website allows the public to search and read the records. The court record summaries provided by WCCA are public records under Wisconsin open records law sections 19.31-19.39 of the Wisconsin Statutes.

The REST interface of WCCA is a subscription based service provided by WCCA to allow automated processes to download and monitor public circuit courts information. We purchased a subscription to this service in May 2020 for the purpose of academic research. We paid $12,500$ USD to the 'Wisconsin Supreme Court' from our research budget, for 12 months of electronic access to WCCA information through the REST interface.

## 3.2 Case Records

Using the REST interface subscription, we were able to download dockets from 1970 to 2020. We queried all case numbers in each county during the period and subsequently, using these case numbers, we queried individual cases for all available information. There are around 11 million records, out of which around 2.5 million are criminal. The case records include the charges in current offense, the outcomes and sentences of cases and the defendants' demographic information (e.g. sex, race, address, and date of birth). There is additional information available about various case events (e.g. hearings, bail decisions, bail amounts etc), information on the associated attorneys and government officials involved, including prosecutors and judges.

## 3.3 Attributes

From these raw case records, we set out to create a curated dataset with numerical and categorical attributes and prediction variable for research. Attributes like **type of offense** (felony, misdemeanor and criminal traffic), **wcisclass** (more specific categorization of offense), **year** of filing, defendant **sex**, and **race** do not require much effort to extract as these are directly available in the case records. As per WCCA declaration (please see the datasheet in appendix F): "In criminal cases, any designation in any race field contains subjective information generally provided by the agency that filed the case." We found that for each case, the defendants are assigned exactly one race. For a small number of defendants, different cases assign a different race to the same defendant. This doesn't necessarily imply mixed-race but could be (and perhaps more likely to be) a data quality issue. More specifically, for $\approx 0.3\%$ of the defendants, there were more than two races reported (across different cases). For another $\approx 0.3\%$ of the defendants, there were two races reported (across different cases) and none of the two races was caucasian. There was perhaps no obvious way to determine the race in these records. These records are not included in the dataset. For $\approx 5\%$ of defendants, there were two races reported (across different cases) and one of the two races was caucasian. We observed that for most of these defendants, non-caucasian race was the most frequently reported race in cases associated with the respective defendant. For reporting model (un)fairness in Section 4.2, we used the non-caucasian race, but a separate column called 'all_cases_races' in the dataset indicates all the races reported for the defendants across cases.

We construct the rest of the attributes and the variable for prediction from the information that is indirectly available in the case records. We use a combination of first name, last name, and date of birth as a unique identifier for a defendant. This identifier allows us to conduct a search in the

database of case records to match the defendant across multiple cases and construct the additional attributes [2]. The detailed process of constructing the additional attributes is described as follows.

### 3.3.1 Prior Criminal Counts

Using the database search, we obtain the prior count of each of the three crime types - felony, misdemeanor and criminal traffic - of the defendant for each of the cases. We were able to collect cases from as early as 1970. We use all these case records for constructing the prior criminal count.

### 3.3.2 Age

We also infer **age at judgment** and **age at first offense** for each case. Age at judgment is calculated based on the date of birth of the defendant and judgment disposition date of the case. Age at first offense for each case is the age when the defendant committed the first crime found in the database.

### 3.3.3 Neighborhood Characteristics

We link 9 local demographics variables to our data from a zipcode level dataset processed from 2010 census data (20). These are **population density**, **proportion who attended college**, **proportion eligible for food stamp**, **African American population share**, **Hispanic population share**, **proportion of male**, **proportions who live in rural and urban areas** and **median household income**. For around 3% of the cases in our dataset, the records didn't contain zipcode. For such cases, we have used average county characteristics. Note that the neighborhood characteristics are from the year 2010, not necessarily from the respective years of individual cases. However, we include the year of filing for each case in the dataset. Researchers can use the year variable to link with external datasets for respective years (not only neighborhood characteristics but other information that they may consider interesting and relevant). Zipcode for the cases can also be made available to the researchers on request for this purpose, for example.

### 3.3.4 Charge Severity

Felonies and misdemeanors are sub-classified by severity level in Wisconsin. Charge severity in raw case records accessed through the REST interface can be Felony A, B, BC, C, D, E, F, G, H, I, U and Misdemeanor A, B, C, U. We map each charge severity to a unique numerical value. The numerical values reflect the increasing charge severity. Mapping is provided in Table 5 in Appendix G. There can be multiple charges in a case; we only keep the highest charge severity for each case.

Using the defendant identifier, we also created the prior counts of charge severity for each prior charge of the defendants. These columns (for each of the 15 charge severity types) are also available in the dataset.

## 3.4 Sentence Length

We calculate the sentence length using the available information about the number of assigned days of jail. In case of multiple charges in the same date, we take the maximum of jail sentence for all charges. If this value is zero, we use the available information about the number of assigned days of house correction (maximum value in case of multiple charges).

For each row, we also provide summary statistics for the previous sentence lengths in four additional columns (max, min, mean, and median sentence length).

### 3.4.1 Other Columns

The dataset contains two other binary variables: **not_detained** and **probation**. `not_detained = 0` implies that the defendant was detained in the case, and `not_detained = 1` implies released, probation or sentence. The other variable, `probation = 1` implies that the defendant got probation.

---

[2]It may be noted that this identifier is not robust to name change by some of the defendants (which we can not rule out). We provide this identifier in the dataset as a separate column.

## 3.5 Recidivism

The choice of the prediction variable in machine learning is a difficult and value-laden decision (21). In this dataset, we provide a commonly used variable in machine learning research, which is whether the defendant recidivates or not. There are well-founded ethical concerns about recidivism prediction task. Please see Section 5.3 for important discussion on ethics. We provide this variable in the dataset to facilitate empirical research on (fundamental) limitations and/or opportunities for improvement. Please see Section 5.2 for discussion on intended uses and limitations. We emphasise that the researchers can use this dataset for other research questions as well and do not have to necessary use the recidivism variable (as a prediction variable or otherwise).

To construct this variable, we had to decide on a follow-up period within which committing a new offense is deemed as recidivism. We pick the usual choice (2; 22) of 2 year follow-up period. Specifically, we measure whether the defendant would commit a crime within two years from the date of judgment.

In any dataset, whether we observe recidivism or not for a defendant is fundamentally biased by the decisions of judges and sentence. This is a common problem (23; 24). Further, defendants serve different sentence lengths and depending on the sentence, it may not be possible to observe recidivism within 2 year follow-up period. Assigning a missing outcome to every case with a sentence throws away a lot of useful data. Yet extending the follow up period for two years after the assigned sentence period instead of the judgment date is also problematic because defendants often serve more or less than the assigned sentence. Since there is not a comparable data source that has the exact jail record of every defendant in Wisconsin, we don't observe the actual (served) sentence length. There is no consensus in the literature about how to deal with this problem. We use a cutoff for sentence length, of 180 days, such that we don't have to throw away a lot of useful data and still leave enough time in the follow-up period for the defendant to reveal crime potential. Above this cutoff, we treat the defendants' outcome as missing, even if they do not re-offend within the follow-up period. As an alternate, we can also define the recidivism variable by varying the sentence length cut-off from 180 days to 2 years, and extending the follow-up period of 2 years by adding the sentence length. Note that the second way takes into account the sentence length given by the judge, which creates even more bias in recidivism variable. Both these recidivism variables are available in our dataset.

## 3.6 Violent Recidivism

In machine learning fairness research, violent recidivism (as opposed to recidivism without any specific criminal context) may also be considered (7; 25). In the FBI's Uniform Crime Reporting (UCR) Program, violent crime is composed of four offenses: murder and nonnegligent manslaughter, forcible rape, robbery, and aggravated assault. In our setting, the challenge in constructing a violent recidivism variable is annotating the large number of records as violent/non-violent. In the raw case records, we have textual descriptions for various charges in a case. There were about $\approx 45,000$ charge descriptions across records, making manual annotation difficult. To the best of our knowledge, there are no publicly available violent/non-violent classifications of these charge descriptions. Note that charge descriptions are different from `type of offense` (felony, misdemeanor and criminal traffic) and `wcisclass` columns. We used GPT-4 (26), a pre-trained large language model, for categorising charge descriptions as referring to violent or non-violent crimes. We provided GPT-4 with the FBI's UCR definition of violent crime in the prompt and asked it to classify a given charge description as violent/non-violent while spelling out the thought process or explanation for the classification. Appendix H provides further details about the process. Due to lack of any ground truth, we do not have a quantitative assessment of the accuracy of the violent/non-violent labels thus obtained. But we did observe several limitations of this approach. We found that GPT often struggled to label charge descriptions that were ambiguous, had insufficient context or contained abbreviations. We also noticed that if prompted multiple times, GPT could classify some charge descriptions differently due to ambiguity and provided plausible explanations for both classifications. We manually chose classifications for a very small fraction of these instances but there can be more such inconsistencies that were not caught. In some other cases, it explicitly stated that due to insufficient information, it could not provide a classification. Further, charge descriptions may have standard meanings in Wisconsin courts which GPT may not have access to in its training data or may not have taken into consideration due to our prompt's focus on FBI's UCR definition. Thus, the process is not perfect and certainly not a replacement for expert annotations.

Using the classifications provided by GPT, we identified whether at least one of the charge descriptions in the case referred to a violent crime. Similar to the recidivism variables in the previous section, the violent recidivism variable is also binary (1 meaning a violent recidivism was observed within a two year follow-up period, with 180 days sentence length cutoff) and contains missing values. Due to space constraints, we do not discuss violent recidivism variable in the rest of this paper, unless explicitly stated. In addition to violent recidivism, another column in the dataset marks whether the current case had a violent charge. We also make available the violent/non-violent classifications for charge descriptions and the explanations provided by GPT as a separate file.

### 3.7 Data Filtering

Through the REST interface, we were able to download case records from as early as 1970, but the records in earlier years tend to be incomplete and the number of cases much smaller. Therefore, we only keep the cases from 2000 in the dataset. It only means that the rows in the curated dataset are only those cases that appear in the courts from 2000. The pre-2000 information for such cases is still included in the form of prior criminal count of defendants. For around 0.005% cases in the dataset, WCCA filing date of older cases is during or after 2000. We have not removed these cases from the dataset. Given the 2 year follow-up period, we exclude cases that are disposed after 2018 since there is not enough time to observe recidivism (we had case records until 2020 only). Note that the post 2018 information is used for observing recidivism outcomes in the dataset. We also point out that during pre-processing, we excluded dismissed cases that do not result in conviction. We also deleted records of defendants that do not have sex and/or race data available or if race could not be determined from the available conflicting race information across cases (see Section 3.3 for details.). We also excluded cases that only have forfeiture (non-crime) charge. Information excluded in pre-processing step is not used in the dataset.

### 3.8 Redacted Information

While the case records on the WCCA website contain personal information of the defendants, the website also displays notice to the employers informing about the violation of state law to discriminate against a job applicant because of an arrest or conviction record. As an ethics consideration on our end, we have removed personal information from the curated dataset that we contribute in this paper. In particular, we have excluded the names of the defendants, their addresses, dates of birth, dates of case events, case numbers etc. We have also replaced information about judge, zipcode and county with pseudo-identifiers, but mappings for zipcode and county can be made available to researchers on request.

## 4 Exploratory Analysis

### 4.1 Summary Statistics

Table 1 presents summary statistics of the dataset thus constructed with around 1.5 million cases from 2000 to 2018. There are five race groups in the dataset, with around 65% Caucasian, 23% African American and 7% Hispanic. The 'Asian or Pacific Islander' and 'American Indian or Alaskan Native' groups are abbreviated as 'Asian' and 'Native American' respectively. The proportion of male defendants in the dataset (80%) is significantly higher than female, and most crimes in the dataset are committed at a younger age (below 60). Misdemeanors are the most frequent crime type except for Hispanic with criminal traffic (45%) being the most common crime type.

Table 2 shows the summary statistics for the two recidivism variables, when the sentence length cut-off is 180 days and 2 years as discussed in Section 3.5. While the rate of missing prediction variable in the dataset varies significantly in the two variables, the recidivism rate (excluding 'missing') doesn't show the same level of significant difference. In the rest of the paper, we use the recidivism variable with 180 days cut-off length, unless otherwise stated. The observed recidivism rate is the highest ($\sim$56%) with Native American.

Figure 1 shows the recidivism rate for different races over the years. Overall recidivism rate and the recidivism rate difference between groups is decreasing over time. We also observed marginal shifts in the distribution of offense types, group proportions, sex, and age. We discuss the limitations of this interpretation in Section 5.2.

Table 1: Summary of the Dataset

|  | Full sample | Caucasian | African American | Hispanic | Native American | Asian |
|---|---|---|---|---|---|---|
| *Sample size* | 1,476,967 | 964,922 | 333,036 | 101,607 | 63,862 | 13,540 |
| *Sample share* |  | 65.33% | 22.55% | 6.88% | 4.32% | 0.92% |
| *Sex* |  |  |  |  |  |  |
|   Male | 80.40% | 79.05% | 83.47% | 88.88% | 69.65% | 87.57% |
| *Age* |  |  |  |  |  |  |
|   Below 30 | 51.38% | 49.45% | 54.13% | 56.91% | 53.71% | 68.60% |
|   30 to 60 | 47.44% | 49.09% | 45.17% | 42.61% | 45.58% | 30.85% |
| *Case type* |  |  |  |  |  |  |
|   Felony | 32.18% | 30.76% | 39.98% | 21.09% | 29.80% | 36.39% |
|   Misdemeanor | 43.04% | 43.67% | 43.14% | 34.12% | 47.55% | 40.89% |
|   Criminal Traffic | 24.78% | 25.57% | 16.88% | 44.79% | 22.66% | 22.73% |

Table 2: Statistics of Two Recidivism Variables

| **Missing Variable Rate** | | | | | | |
|---|---|---|---|---|---|---|
| Sentence Cut-off | Overall | Caucasian | African American | Hispanic | Native American | Asian |
| 180 Days | 0.0807 | 0.0667 | 0.1286 | 0.0653 | 0.0686 | 0.0767 |
| 2 Years | 0.0471 | 0.0367 | 0.0824 | 0.0365 | 0.0374 | 0.0513 |
| **Recidivism Rate (excluding 'missing')** | | | | | | |
| Sentence Cut-off | Full sample | Caucasian | African American | Hispanic | Native American | Asian |
| 180 Days | 0.4221 | 0.4034 | 0.4643 | 0.3876 | 0.5647 | 0.3780 |
| 2 Years | 0.4168 | 0.3996 | 0.4534 | 0.3825 | 0.5593 | 0.3724 |

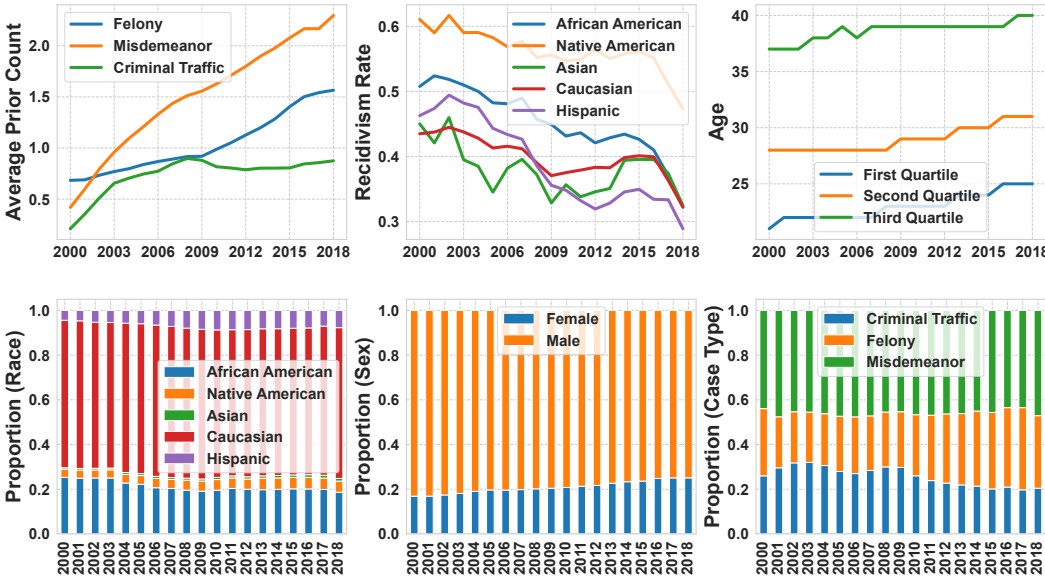

Figure 1: Change in Data Distribution with Time

For each type of offense in the dataset, there are more specific categorization of offense in the dataset under the `wcisclass` variable. For misdemeanor, there are 55 distinct `wcisclass` values (and 0.33% missing). For felony, there are 60 distinct `wcisclass` values (and 0.34% missing), and for criminal traffic, there are 39 distinct `wcisclass` values (and 0.49% missing). We list the 10 most common `wcisclass`s for each offense type in Table 3. We provide summary statistics about `highest_charge_severity` in Table 6 in appendix G. Figure 2 in appendix G also compares the

distributions of different offense types in the two recidivism variables, when the recidivism value is missing and when it is not not-missing.

Table 3: 10 Most Frequent `wcisclass` for Each Type of Offense

| Misdemeanor | | Felony | | Criminal Traffic | |
|---|---|---|---|---|---|
| `wcisclass` | Percentage | `wcisclass` | Percentage | `wcisclass` | Percentage |
| Disorderly Conduct | 15.51 | Drug Possession | 15.98 | OAR/OAS | 42.74 |
| Battery | 14.44 | Bail Jumping | 10.43 | Operating While Intoxicated | 42.25 |
| Resisting Officer | 13.62 | Burglary | 7.49 | Operate Without License | 8.37 |
| Drug Possession | 8.76 | Drug Manufacture/ Deliver | 6.94 | Hit and Run | 1.98 |
| Bail Jumping | 8.75 | Theft | 6.28 | Unidentified Misdemeanor Traffic | 1.16 |
| Theft | 6.31 | Operating while intoxicated | 5.83 | Bail Jumping | 0.9 |
| Retail Theft (Shoplifting) | 5.62 | Other Felony | 5.03 | Other Misdemeanor | 0.78 |
| Criminal Damage | 5.61 | Forgery | 3.76 | nan | 0.49 |
| Drug Paraphernalia | 2.96 | Battery | 3.25 | Resisting Officer | 0.41 |
| Worthless Checks | 2.69 | Substantial/ Aggravated Battery | 3.06 | Drug Possession | 0.21 |

## 4.2 Machine Learning Predictions

The purpose of this section is not to indicate that the following is the "right" way or the only way to use this dataset in research. But for reference only, we report performance metrics of two machine learning classifiers on this dataset. The predictors included in the models were sex, type of offense, prior criminal count (for each type), and age (at judgment and at first offense). Logistic Regression and XGBoost classifiers were from Python's scikit-learn (27) and XGBoost (28) packages respectively. The packages are public and freely available. LR learns a linear model while XGBoost learns a decision tree ensemble. We split the entire data into 70% train and 30% test, and the results are reported on the test data. For LR classifier, we include L2 regularization and select the regularization parameter via 10-fold cross-validation. For XGBoost, we include both L1 and L2 regularization and tune the hyperparameters via grid search and 5-fold cross-validation. 95% confidence intervals are constructed by multiple train and test splits. Details of compute resources can be found at `https://scicomp.ethz.ch/wiki/Euler#Introduction`.

Table 4 shows the performance of the LR classifier and the XGBoost classifier on this dataset across all five race groups. FPR stands for False Positive Rate, FNR for False Negative Rate, PR for the rate of positive classification, AUC stands for Area under the ROC Curve. We observe from the table that Native American is the most disadvantaged group in the sense that it has the highest FPR and PR, and the lowest FNR. The largest FPR difference is around 13%, the largest FNR difference is around 18%, and the largest PR difference is around 20%, all between Native American and Hispanic with XGBoost classifier. Hispanic and Caucasian groups receive the most favorable decisions (1-PR) in both models, followed by Asian and African American groups. The overall accuracy of XGBoost and LR is not very different but the FNR, FPR and PR differences between groups is higher for XGBoost. These results are for the recidivism variable with 180 days sentence cut-off. In Table 7 in appendix G, we also provide results for the recidivism variable with 2 years sentence cut-off. We did not observe a notable difference in performance metrics for the two recidivism variables but it may be worth investigating the differences at a more granular level.

In addition to the discussion in Section 2 on the limitations of the COMPAS dataset, it is useful to consider the following as well for comparison with our dataset. (29) report about COMPAS/ProPublica

dataset: "The (unconstrained) logistic regression classifier leads to an accuracy of 0.668. However, the classifier yields false positive rates of 0.35 and 0.17, respectively, for blacks and whites (i.e., DFPR = 0.18), and false negative rates of 0.31 and 0.61 (i.e., DFNR = 0.30)." The COMPAS dataset has the following number of samples from different racial groups. African-American: 3175, Asian: 31, Caucasian: 2103, Hispanic: 509, Native American: 11. Due to very small numbers, racial groups other than African-American and Caucasian are generally not analysed in the literature using the COMPAS. Our dataset significantly addresses that limitation.

Table 4: Recidivism Prediction Performance on the Dataset, with 95% Confidence Intervals

|  | Overall | Caucasian | African American |
|---|---|---|---|
| *XGBoost* | | | |
| Accuracy | $0.6588 \pm 0.0018$ | $0.6648 \pm 0.0020$ | $0.6459 \pm 0.0034$ |
| AUC | $0.7039 \pm 0.0018$ | $0.7044 \pm 0.0023$ | $0.7033 \pm 0.0035$ |
| FPR | $0.2244 \pm 0.0041$ | $0.2159 \pm 0.0042$ | $0.2454 \pm 0.0048$ |
| FNR | $0.5008 \pm 0.0040$ | $0.5113 \pm 0.0045$ | $0.4792 \pm 0.0067$ |
| PR | $0.3405 \pm 0.0035$ | $0.3261 \pm 0.0038$ | $0.3734 \pm 0.0041$ |
| *Logistic Regression* | | | |
| Accuracy | $0.6461 \pm 0.0013$ | $0.6560 \pm 0.0020$ | $0.6206 \pm 0.0027$ |
| AUC | $0.6822 \pm 0.0013$ | $0.6825 \pm 0.0022$ | $0.6806 \pm 0.0030$ |
| FPR | $0.1532 \pm 0.0022$ | $0.1479 \pm 0.0026$ | $0.1667 \pm 0.0031$ |
| FNR | $0.6282 \pm 0.0029$ | $0.6334 \pm 0.0035$ | $0.6244 \pm 0.0039$ |
| PR | $0.2456 \pm 0.0021$ | $0.2363 \pm 0.0024$ | $0.2638 \pm 0.0023$ |
|  | Hispanic | Native American | Asian |
| *XGBoost* | | | |
| Accuracy | $0.6567 \pm 0.0055$ | $0.6303 \pm 0.0076$ | $0.6760 \pm 0.0108$ |
| AUC | $0.6719 \pm 0.0068$ | $0.6878 \pm 0.0090$ | $0.7079 \pm 0.0122$ |
| FPR | $0.2016 \pm 0.0073$ | $0.3272 \pm 0.0145$ | $0.2203 \pm 0.0147$ |
| FNR | $0.5674 \pm 0.0096$ | $0.4025 \pm 0.0106$ | $0.4944 \pm 0.0225$ |
| PR | $0.2910 \pm 0.0060$ | $0.4800 \pm 0.0100$ | $0.3283 \pm 0.0128$ |
| *Logistic Regression* | | | |
| Accuracy | $0.6535 \pm 0.0048$ | $0.6026 \pm 0.0057$ | $0.6735 \pm 0.0116$ |
| AUC | $0.6558 \pm 0.0056$ | $0.6746 \pm 0.0068$ | $0.6946 \pm 0.0136$ |
| FPR | $0.1292 \pm 0.0051$ | $0.2395 \pm 0.0081$ | $0.1386 \pm 0.0118$ |
| FNR | $0.6903 \pm 0.0085$ | $0.5189 \pm 0.0080$ | $0.6350 \pm 0.0199$ |
| PR | $0.1991 \pm 0.0049$ | $0.3761 \pm 0.0065$ | $0.2243 \pm 0.0106$ |

# 5 Discussion

## 5.1 License

The dataset is distributed under the Creative Commons 4.0 BY-NC-SA (Attribution-NonCommercial-ShareAlike) license. The choice of this license is motivated by the intended uses of the dataset and other considerations that we discuss next.

## 5.2 Intended Uses and Limitations

The dataset is intended to be used for academic research only. We focus in this paper on algorithmic fairness analysis as a specific use-case, but it doesn't have to restricted to this. We have provided pseudo-identifiers for judge, county and zipcode and as stated earlier, mappings for county and zipcode can be provided to researchers on request to explore different research use-cases or to explore the interaction of algorithmic or data centric factors with systemic issues such as biases of judges, county or location factors etc. There is more information available in the raw case records that is more difficult to parse and thus, not available at present in the curated dataset. As we discuss in Section 5.4, we will strive to update the dataset in the near future if we find this additional information to be of sufficiently good quality. This will further extend the scope of research uses of the dataset. These

may include, for e.g., studying the process of sentencing itself, or measuring the impact of defendant attributes on the severeness of sentencing or bail.

The dataset addresses several limitations of prior datasets (larger size, large number of samples from different racial groups, different courts, counties, more attributes, less variance, observations over longer time etc). At the same time, there are several fundamental limitations (11) that are difficult or even impossible to address in any dataset. Examples include missing values and biases encoded in the recidivism variable, as discussed in Section 3.5, and in the attributes. Such biases in the data can not be avoided. Depending on the context, they may be handled in other ways, for e.g., through appropriate algorithmic intervention or even critically rethinking the use of algorithms in the first place. We argue that since these limitations are fundamental and can not be addressed easily through data collection or curation processes, it is important that the datasets are available to researchers for critical research. The researchers must be aware of the limitations of the datasets however. For example, the unavailability of prison/jail records implies that the reported recidivism is not guaranteed to be accurate, and we are not able to measure the extent of this inaccuracy. In addition, since we only have cases from Wisconsin Circuit Courts, we can not capture recidivism elsewhere, also, an error that is hard to quantify the impact of. Moreover, the fact that convictions do not faithfully represent criminality is another source of bias and a fundamental issue with the prediction task definition itself. We further bring attention to the fact that the WCLD dataset is built using case records for a long period of time. There are several changes in the society and external variables over such a long time period that can not be captured in a dataset. This temporal factor must be kept in mind while using the dataset for research. However, it is also a strength of the dataset since it can be used to study the algorithms' behavior under distributions shifts. We note that not all distribution shifts are reflective of the real-world societal changes but are nevertheless realities of the data collection and curation processes, and therefore, availability of such datasets for research is important. In case of WCCA, case records from earlier years like late 1900s tend to be less in number (there can be multiple possible reasons for this), and that may have caused underestimation of prior criminal counts for older cases in the WCLD dataset.

As part of the subscription information, WCCA also informed of a few other limitations of the data available through the REST interface. We list those limitations in the datasheet (Section F.3 in appendix F).

## 5.3 Ethical Considerations

The court record summaries provided by WCCA are public records under Wisconsin open records law sections 19.31-19.39 of the Wisconsin Statutes. Court records not open to public inspection by law are not available. WCCA information does not include information that may be confidential, sealed, or redacted in accordance with all applicable statutes, court orders, and rules related to confidentiality, sealing, and redaction. We refer the reader to the WCCA FAQs (30) for further information. We took additional steps to redact sensitive information as discussed in Section 3.8

Further, readers should not interpret summary statistics of this dataset (for example, those listed in Tables 1 and 2) as 'ground truth' but rather as characteristics of the dataset only.

Researchers using this dataset are expected to carefully take into account the intended uses and limitations listed in Section 5.2 and follow practices for responsible use of criminal justice data in research (11). Beyond limitations of our dataset and data more broadly in this context, fundamental limitations of algorithms (including but not limited to mathematical limitations (3; 4)) pose challenge to addressing the ethical concerns (11; 31; 32) related to recidivism prediction task itself (33). Such limitations and ethical concerns about the task should also be taken into account while drawing conclusions from the research using this dataset.

## 5.4 Future Work

In the case records that we downloaded using the REST interface, there is additional information available about various case events (e.g. hearings, bail decisions, bail amounts, sentences), information on the associated attorneys and government officials involved, including prosecutors. It may be useful to parse this information and update the dataset. We will strive to do these and update the dataset depending on the quality and usefulness of the additional information available. Readers interested in contributing to the dataset are encouraged to reach out to us.

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

# F    Datasheet

## F.1    Motivation

**For what purpose was the dataset created?** The raw case records were created presumably for public inspection and for record keeping in the courts.

The external census dataset used in curation was created for census.

The final curated dataset, contributed in this paper, was created for academic research on algorithmic fairness.

**Who created the dataset, and on behalf of which entity?** According to Wisconsin Circuit Courts Access website, the raw data records that we accessed through their website are an exact copy of the case information entered into the circuit court case management system by court staff in the counties where the case files are located.

Nianyun Li, Claudia Marangon and Peiyao Sun curated the dataset in its current form with the help of Elliott Ash and Naman Goel.

**Who funded the creation of the dataset?** The funding of creation of original data records is unclear, but presumably the state funded the court staff in the counties where the case files are located.

The creation of the curated dataset was funded by ETH Zurich.

## F.2    Composition

**What do the instances that comprise the dataset represent?** The instances represent case and defendant information.

**How many instances are there in total (of each type)?** There are around 1.5 million instances.

**Does the dataset contain all possible instances or is it a sample of instances from a larger set?** The dataset is a sample of instances from a larger set.

**What data does each instance consist of?** Each instance contains defendant's new_id, age, sex, type of offense, wcisclass, year of filing, race, age at judgment, age at first offense, 9 neighborhood characteristics (population density, proportion who attended college, proportion eligible for food stamp, African American population share, Hispanic population share, proportion who live in rural and urban area, median household income, highest charge severity, not_detained, probation, recid (180 days sentence length cutoff), recid (2 years sentence length cutoff), violent_recid (180 days sentence length cutoff), jail, county, violent_crime. The details of each of these variables are provided in the accompanying paper and the metadata file in the data directory. Each instance also contains prior criminal count for each type of offense, prior sentence length statistics, prior charge severity counts. The counts were created from the **available** raw case records by performing database search, and are therefore possibly underestimated.

**Is there a label or target associated with each instance?** For research, recid (180 days sentence length cutoff) and recid (2 years sentence length cutoff) are two variables associated with each instance. There is also a violent_recid (180 days sentence length cutoff) variable available for research. These are not ground truth labels in a traditional sense but only variables defined by the authors. Details in Section 3.5and ethics discussion in Section 5.3 of the accompanying paper.The first variable is recidivism as observed in the case records by performing database search, within a 2 year follow-up period since judgment disposition date, using a 180 days cut-off for sentence. The second is obtained by using a 2 year cut-off sentence and extending the follow-up period of 2 years by adding the sentence length.

**Is any information missing from individual instances?** Yes, the recidivism variables can not be observed for defendants depending on their sentence. Thus, it is missing for some defendants. Details in Section 3.5.

**Are relationships between individual instances made explicit?** The dataset is anonymized and therefore some relationship between individual instances may be lost. We have included defendant pseudo-identifier in the dataset constructed based on first name, last name and date of birth.

**Are there recommended data splits?** No. But we have included two possible random splits in the dataset (one is completely random thus only ensuring different cases in train and test splits, the other also ensures different defendants in train and test splits).

**Are there any errors, sources of noise, or redundancies in the dataset?** The errors are possible in the raw case information entered by the court staff. Known errors in the curated dataset construction are discussed in Section 5.2.

**Is the dataset self-contained, or does it rely on external resources?** The dataset has been curated using case records from WCCA and census data from 2010. The curated dataset is self-contained.

**Does the dataset contain data that might be considered confidential?** No

**Does the dataset contain data that, if viewed directly, might be offensive, insulting, or threatening?** No

**Does the dataset relate to people?** Yes, the raw case records relate to the defendants. However, directly identifiable information such as names, addresses, date of birth, case numbers etc has been removed in the curated dataset.

**Does the dataset identify any subpopulations?** Yes, the dataset contains data from five racial groups as marked by WCCA, sex and age groups.

**Is it possible to identify individuals?** The raw case records available on WCCA are public information and it is possible to identify individuals there. For the curated dataset that we release, we have removed directly identifiable information such as names, addresses, date of birth, case numbers etc.

**Does the dataset contain data that might be considered sensitive in any way?** The raw case records available on WCCA are public information and some of the information such as defendant's personal information may be considered sensitive. For the curated dataset that we release, we have removed directly identifiable information such as names, addresses, date of birth, case numbers etc.

### F.3    Collection Process

**How was the data associated with each instance acquired?** Through the REST interface of WCCA.

**What mechanisms or procedures were used to collect the data?** Through the REST interface of WCCA. We queried all case numbers in each county during the period and subsequently, using these case numbers, we queried individual cases for all available information. Different attributes were then derived using the process described in the accompanying paper.

**If the data are a sample from a larger set, what was the sampling strategy?** The dataset is composed primarily of new cases filed between 2000-2018. The dataset excludes dismissed cases that do not result in conviction, records of defendants that do not have sex and/or race data and cases that only have forfeiture (non-crime) charge. `https://wcca.wicourts.gov/faq.html` provides more information on records that might have been deleted. WCCA also informed us of a few limitations of the data as part of the subscription. These are listed as follows:

1. WCCA Information includes only court records open to public view under Wisconsin's Open Records Law, Wis. Stat. 19.31-19.39. Court records not open to public inspection by law are not available.

2. WCCA Information does not include information that may be confidential, sealed, or redacted in accordance with all applicable statutes, court orders, and rules related to confidentiality, sealing, and redaction.

3. WCCA Information consists of information entered into the CCAP [3] case management system by the Clerk of Circuit Court or Register in Probate in each county. CCAP is not responsible for the accuracy or timeliness of WCCA information.

4. WCCA Information does not comprise the complete court record. Copies of documents must be obtained from the Clerk of Circuit Court or Register in Probate.

5. WCCA Information is only a snapshot of the information accessible in the CCAP case management system on the date the information is downloaded by the Subscriber.

---

[3]WCCA was formerly CCAP.

6. WCCA Information is not the Judgment and Lien Docket under Wis. Stat. 806.10. The Judgment and Lien Docket is available from the Clerk of Circuit Court.

7. Court records which predate the implementation of the CCAP case management system in the county in which the records were created are not accessible under this Agreement, except to the extent such records have been back loaded.

8. In criminal cases, any designation in any race field contains subjective information generally provided by the agency that filed the case.

9. Searching WCCA Information by a particular field or code may not return all cases in which a particular event occurred unless at the time the record was created the case management system required the field or code to be completed in order to proceed to make the rest of the record

**Who was involved in the data collection process and how were they compensated?** The case records were created by court staff in respective county courts and were presumably, compensated by state.

The authors of this paper collected the case records from WCCA and were employees of ETH Zurich during the data curation process. They were compensated by ETH Zurich in the form of fixed monthly salaries. Nianyun Li, Naman Goel, Peiyao Sun, Claudia Marangon were/are on fixed-term contracts with ETH Zurich.

**Over what time frame was the data collected?** Authors had access to the raw case records through WCCA REST interface during the period July 2020 - July 2021. However, data collection was finished by Feb 2021.

**Were any ethical review processes conducted?** The authors are not aware of the ethical review process followed in WCCA or county courts for creation of the case records. As part of the curation of the dataset that we contribute, no formal/institutional ethical review process was conducted.

**Does the dataset relate to people?** Yes, the raw case records relate to defendants. However, directly identifiable information such as names, addresses, date of birth, case numbers etc has been removed in the curated dataset.

**Did you collect the data from the individuals directly, or obtain it via third parties?** We obtained the raw case records from a third party, WCCA (`https://wcca.wicourts.gov`).

**Were the individuals notified about the data collection?** The authors are not aware of it. However, presumably, the defendants were aware that the information about their cases (and hence the related information about them) is kept in court records and is public information under Wisconsin state laws, when exceptions do not apply.

**Did the individuals in question consent to the collection and use of their data?** The authors are not aware of it. The information is available publicly under Wisconsin state laws.

**If consent was obtained, were the consenting individuals provided with a mechanism to revoke their consent in the future or for certain uses?** WCCA provides option in certain limited cases to petition to have case-records removed. Please see `https://wcca.wicourts.gov/faq.html`.

For the curated dataset, directly identifiable information such as names, addresses, date of birth, case numbers etc has already been removed.

**Has analysis of the potential impact of the dataset and its use on data subjects been conducted?** Authors are not aware if Wisconsin state or WCCA had done any such analysis.

For the curated dataset, we have thought carefully about it and redacted all the information that, according to us, could potentially affect subjects.

### F.4  Pre-processing and Cleaning

**Was any preprocessing of the data done?** The accompanying paper describes the details of curating the dataset from the raw case records.

**Was the "raw" data saved in addition to the cleaned data?** Yes, we saved the data on our institute's secure servers (until our research requires). We do not plan to make this data available to others.

**Is the software used to clean the data available?** We didn't use any 'data cleaning software'. We have described the steps taken in curating the dataset in the accompanying paper. If useful, we can also provide SQL commands for these steps.

## F.5 Uses

**Has the dataset been used for any tasks already?** The curated dataset that we release has only been used for academic research. We are not aware who else has used the raw case records from WCCA and for which tasks.

**Is there a repository that links to any or all papers that use the dataset?** Not to our knowledge.

**What (other) tasks could the dataset be used for?** The dataset is for academic research.

**Is there anything about the composition of the dataset or the way it was collected and cleaned that might impact future uses?** We have listed limitations of the data in Section 5.2 and in earlier parts of this datasheet (for example, see Section F.3).

**Are there tasks for which the dataset should not be used?** The dataset should not be used for purposes other than academic research.

## F.6 Distribution

**Will the dataset be distributed to third parties outside of the entity on behalf of which the dataset was created?** Yes, public data.

**How will the dataset be distributed?** The dataset is hosted at `http://clezdata.github.io/wcld/`. Downloads are subject to research only use acknowledgement. In case of any difficulties in accessing the data in the future, interested readers can contact the authors.

**When will the dataset be distributed?** The dataset is hosted at `http://clezdata.github.io/wcld/`.

**Will the dataset be distributed under a copyright, other IP license, or terms of use?** The dataset is distributed under the Creative Commons 4.0 BY-NC-SA license.

**Have any third parties imposed IP-based or other restrictions on the data associated with the instances?** No.

**Do any export controls or other regulatory restrictions apply to the data?** No.

## F.7 Maintenance

**Who is supporting/hosting/maintaining the dataset?** Elliott Ash.

**How can the data owner/curator be contacted?** Through email: elliott.ash@gess.ethz.ch

**Is there an erratum?** Not at the time of publishing this paper.

**Will the dataset be updated?** The existing entries in the dataset are unlikely to be modified. New information may be added.

**If the dataset relates to people, are there applicable limits on the retention of data associated with the instances?** Public information under applicable law.

**Will older versions of the dataset continue to be supported/hosted/maintained?** In the unlikely event that the entries in the dataset are to be modified, older version will also be made available, for example, using a version control system.

**If others want to extend/augment/build on/contribute to the dataset, is there a mechanism for them to do so?** Individuals interested in contributing are encouraged to contact Elliott Ash at elliott.ash@gess.ethz.ch.

# G   Additional Information

Table 5 shows the mappings from charge severity categories on WCCA and the numerical ranking that we assigned to these categories. Values 1-6 were assigned for forfeiture charges and hence, not shown in the table. Table 6 provides summary statistics for the column charge severity.

Table 5: Mapping from Charge Severity to Numerical Ranking

| Charge | highest_charge_severity |
|---|---|
| Felony A | 21 |
| Felony B | 20 |
| Felony BC | 19 |
| Felony C | 18 |
| Felony D | 17 |
| Felony E | 16 |
| Felony F | 15 |
| Felony G | 14 |
| Felony H | 13 |
| Felony I | 12 |
| Felony U | 11 |
| Misdemeanor A | 10 |
| Misdemeanor B | 9 |
| Misdemeanor C | 8 |
| Misdemeanor U | 7 |

Table 6: Highest Charge Severity in the Dataset

| highest_charge_severity | Count | Percentage |
|---|---|---|
| 7 | 516004 | 34.94 |
| 10 | 460898 | 31.21 |
| 9 | 145750 | 9.87 |
| 13 | 116008 | 7.85 |
| 12 | 74062 | 5.01 |
| 15 | 37580 | 2.54 |
| 14 | 30561 | 2.07 |
| 18 | 26678 | 1.81 |
| 11 | 21893 | 1.48 |
| 16 | 21217 | 1.44 |
| 17 | 17088 | 1.16 |
| 20 | 6187 | 0.42 |
| 19 | 1581 | 0.11 |
| 21 | 845 | 0.06 |
| 8 | 615 | 0.04 |

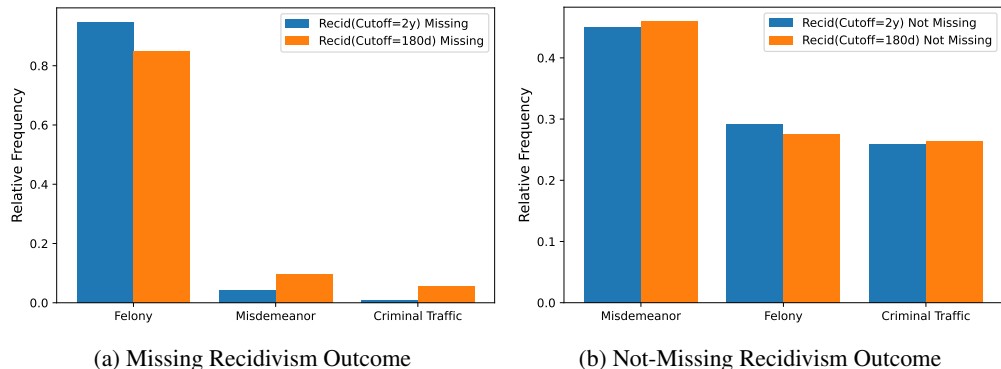

(a) Missing Recidivism Outcome    (b) Not-Missing Recidivism Outcome

Figure 2: Differences in the distribution of type of offense depending on the sentence cutoff length in recidivism variables.

Table 7 shows results of the machine learning classifiers for the recidivism variable with 2 years sentence cut-off.

Table 7: Recidivism (2 year Sentence Cut-Off) Prediction with 95% Confidence Intervals

| | Overall | Caucasian | African American |
|---|---|---|---|
| *XGBoost* | | | |
| Accuracy | $0.6589 \pm 0.0016$ | $0.6652 \pm 0.0019$ | $0.6460 \pm 0.0038$ |
| AUC | $0.7018 \pm 0.0018$ | $0.7026 \pm 0.0021$ | $0.7005 \pm 0.0034$ |
| FPR | $0.2177 \pm 0.0029$ | $0.2094 \pm 0.0029$ | $0.2353 \pm 0.0046$ |
| FNR | $0.5136 \pm 0.0043$ | $0.5232 \pm 0.0049$ | $0.4970 \pm 0.0064$ |
| PR | $0.3298 \pm 0.0030$ | $0.3163 \pm 0.0032$ | $0.3567 \pm 0.0039$ |
| *Logistic Regression* | | | |
| Accuracy | $0.6457 \pm 0.0014$ | $0.6555 \pm 0.0016$ | $0.6215 \pm 0.0029$ |
| AUC | $0.6784 \pm 0.0015$ | $0.6794 \pm 0.0019$ | $0.6760 \pm 0.0034$ |
| FPR | $0.1478 \pm 0.0022$ | $0.1430 \pm 0.0023$ | $0.1571 \pm 0.0026$ |
| FNR | $0.6429 \pm 0.0024$ | $0.6469 \pm 0.0030$ | $0.6453 \pm 0.0047$ |
| PR | $0.2351 \pm 0.0018$ | $0.2270 \pm 0.0020$ | $0.2467 \pm 0.0028$ |
| | Hispanic | Native American | Asian |
| *XGBoost* | | | |
| Accuracy | $0.6576 \pm 0.0054$ | $0.6273 \pm 0.0064$ | $0.6733 \pm 0.0149$ |
| AUC | $0.6724 \pm 0.0056$ | $0.6852 \pm 0.0066$ | $0.7016 \pm 0.0182$ |
| FPR | $0.1997 \pm 0.0079$ | $0.3223 \pm 0.0123$ | $0.2153 \pm 0.0130$ |
| FNR | $0.5720 \pm 0.0097$ | $0.4125 \pm 0.0105$ | $0.5140 \pm 0.0245$ |
| PR | $0.2872 \pm 0.0076$ | $0.4706 \pm 0.0091$ | $0.3163 \pm 0.0116$ |
| *Logistic Regression* | | | |
| Accuracy | $0.6540 \pm 0.0054$ | $0.5988 \pm 0.0050$ | $0.6728 \pm 0.0152$ |
| AUC | $0.6533 \pm 0.0061$ | $0.6699 \pm 0.0070$ | $0.6878 \pm 0.0176$ |
| FPR | $0.1293 \pm 0.0058$ | $0.2358 \pm 0.0090$ | $0.1339 \pm 0.0113$ |
| FNR | $0.6948 \pm 0.0087$ | $0.5316 \pm 0.0082$ | $0.6521 \pm 0.0222$ |
| PR | $0.1967 \pm 0.0055$ | $0.3658 \pm 0.0069$ | $0.2138 \pm 0.0096$ |

# H    Violent/Non-Violent Labels for Charge Descriptions using GPT-4

The prompt used for GPT was as follows:

"In the FBI's Uniform Crime Reporting (UCR) Program, violent crime is composed of four offenses: murder and nonnegligent manslaughter, forcible rape, robbery, and aggravated assault. Violent crimes are defined in the UCR Program as those offenses which involve force or threat of force.
I will next provide you a list of charge descriptions. For each charge description in the list, I want you to provide me a single word answer (Violent/Non-Violent), depending on whether that charge description refers to a violent crime or not. Before providing the answer, I want you to provide an explanation or thought process of how you go from charge description to the answer. The format of your response should be Charge Description;;Thought Process;;Violent/Non-Violent. Do not include any other text in your response. The charge description in your response should be exactly same as the charge description in my list (do not correct the formatting or spellings etc in charge descriptions) and in the same order as my list.
Here is the list: "

We appended charge descriptions (50 at a time) to the prompt and repeatedly prompted the model until we obtained labels for all the charge descriptions. The model used was 'gpt-4' as on 20-Sep-2023. We set the system message as 'You are a helpful assistant.' The cost of inference was approximately $150, including the costs of trial and error with different prompting and models etc.

