# OpenReview forum: "WCLD: Curated Large Dataset of Criminal Cases from Wisconsin Circuit Courts"
_NeurIPS.cc/2023/Track/Datasets_and_Benchmarks — NeurIPS 2023 Datasets and Benchmarks Poster_

### Official Review · Reviewer_qVAQ · 2023-06-30

**Rating:** 7
**Confidence:** 4

**Strengths:**

The WCLD dataset will be valuable to the fairness community, in which many researchers have focused on criminal sentencing/recidivism and there is a large body of literature on this setting. I think in particular, the WCLD provides derived features and recidivism outcomes like the COMPAS dataset, but is much larger and more diverse in case type than the COMPAS dataset.

In terms of ethical and social implications, the authors discuss various considerations for deriving the recidivism target variable in Section 3.5. They carefully consider the interaction between sentence lengths and the standard time cut-off for the follow-up period for which committing a new offense is considered recidivism. They also discuss steps taken to filter the data for quality control and steps taken to redact sensitive information about the defendant/judges in Section 3.6 and 3.7.

Overall, I found the documentation for the dataset to also be exceptionally clear and comprehensive.


**Additional Feedback:**

Small edits

Page 2
Line 70: datsets -> datasets

Page 9
Line 264: “doesn’t have to restricted” -> “doesn’t have to be restricted”
Line 289: less -> fewer
Line 294: Capitalize “appendix”

**Clarity:**

Yes, the paper clearly outlines how the WCLD dataset differs from existing criminal justice datasets, how the dataset was created and various design choices, how the dataset will be hosted/licensed/maintained, and intended use cases/limitations.

**Correctness:**

Yes, the dataset is filtered in a reasonable and consistent way with respect to the derived features. The XGBoost and logistic regression models are reasonable choices for baseline models for the binary classification task derived from the dataset. The authors perform hyperparameter search with 5 fold cross-validation and report results with uncertainty over multiple splits.

**Documentation:**

Yes, Section 3 on Data Curation clearly explains how the dataset was created. I believe I could recreate the dataset from the raw case records with this information.

The additional checklist points to all of the required documentation. I used the URL provided to access the dataset. Section 5.2. details intended use cases and limitations. Section 5.1. details the license for the dataset. Appendix C provides a datasheet with more detailed information about dataset creation, intended uses, limitations, hosting, and maintenance.


**Ethics:**

No, the protection of defendant/judge identities in the dataset is an ethically relevant issue, but I think the authors clearly explain how they redacted/anonymized the data and follow standard best practices to my knowledge. I think they actually go beyond what is strictly necessary to ensure a stronger guarantee on privacy.

**Limitations:**

From my understanding of how the recidivism outcome is derived, the authors take two approaches. One that (a) applies a shorter cut-off to the sentence length of 180 days to ensure there is adequate time between the end of the sentence and the 2 year follow-up period from the date of judgment and does not report recidivism outcomes for cases with longer sentence lengths and another that (b) applies a cut-off to the sentence length of 2 years and sets the 2 year follow-up period to after the end of the sentence length. The authors note that the second approach introduces bias by taking into account the sentence length, but I’d like to see a more systematic analysis of how these design choice affects the types of cases retained in the dataset and considered cases where defendants recidivate.

**Opportunities For Improvement:**

The derived features are explained in Section 3, but I think the actual dataset file could benefit from an additional schema/key for the column names, what feature they correspond to, and what type of data they contain (continuous, discrete, pseudo-identifier, etc.).

The 2 year follow-up period within which committing a new offense is considered recidivism is a standard set by prior work, but it would be helpful to include context on why this is the standard. In particular, this feels like it would be a short follow-up period for more serious crimes, where the sentence length is long or perhaps the lead up time to committing a new serious offense is longer. Similarly, with a sentence length cut-off of 180 days, data for very serious crimes would be excluded. Instead of assessing whether setting different sentence length cut-offs affects model performance, as you explore in Section 4.2., it might be worthwhile to perform a more descriptive analysis of how the distribution over crime type/charge severity changes for the cases you treat as having missing defendants’ outcomes (sentence length above the cut-off threshold) as you increase the sentence length cut-off or how the distribution over crime type/charge severity changes for the cases where defendants recidivate.

The authors mention at a couple points in the paper that the COMPAS dataset may not be an ideal benchmark for fairness evaluations/interventions due to its low variance. I’m assuming that this refers to the variance of feature values in the dataset, but I wonder if there is a way to quantify this for the COMPAS vs. WCLD dataset, or at least for the features used as input to the models evaluated. Relatedly, I think one of the weaker sections of the paper is Section 4.2. I think it could be strengthened with model evaluations on COMPAS that could enable a comparison between the two datasets or even fairness interventions on both datasets.



**Relation To Prior Work:**

Yes, the Related Work section clearly discusses how WCLD differs from COMPAS (larger, more diverse) and other existing criminal justice datasets (many of which are raw court documents instead of tabular data with recidivism outcomes).

**Summary And Contributions:**

This paper introduces a recidivism dataset, WCLD, derived from publicly accessible criminal case records from circuit courts in the state of Wisconsin. The dataset has some similar features to the well-known COMPAS dataset, derived from criminal case records from one court in Broward County, Florida. Compared to the COMPAS dataset, which contains 7000 rows from one court, the WCLD dataset contains 1.5 million rows from circuit courts across the state of Wisconsin. The authors construct the dataset by accessing case records through the Wisconsin Circuit Courts Access (WCCA) API and derive additional features such as prior criminal counts, sentence length, etc. from these raw case records and linking to census data.

---

> ### Author Response · Authors · 2023-08-25
> **Response (Reviewer qVAQ)**
>
> We thank the reviewer for their time and useful suggestions.
>
> 1. *"... descriptive analysis of how the distribution over crime type/charge severity changes for the cases we treat as having missing defendants’ outcomes ... as we increase the sentence length cut-off  ...."*\
> \
> Since images are perhaps not supported on OpenReview, we include the relative frequency numbers below. There are some interesting observations to be made here. For e.g., case type distribution for missing outcomes is different in the two cutoff lengths, but the change seems less noticeable for non-missing outcomes. However, we note that the number of cases with missing outcomes is much smaller than non-missing outcomes. We agree with the reviewer that more such interesting analysis can be performed on this dataset by researchers in the future.\
> \
> Recid(Cutoff=2y) Missing, Case Type Relative Frequencies\
> [('Felony', 0.9478197173388486), ('Misdemeanor', 0.04285878432724348), ('Criminal Traffic', 0.009321498333907849)]\
> \
> Recid(Cutoff=180d) Missing, Case Type Relative Frequencies\
> [('Felony', 0.8485639029627889), ('Misdemeanor', 0.09517498238432373), ('Criminal Traffic', 0.056261114652887294)]\
> \
> Recid(Cutoff=2y) Not Missing, Case Type Relative Frequencies\
> [('Misdemeanor', 0.4495592048278209), ('Felony', 0.2908288881956993), ('Criminal Traffic', 0.2596119069764798)]\
> \
> Recid(Cutoff=180d) Not Missing, Case Type Relative Frequencies
> [('Misdemeanor', 0.4598193341213989), ('Felony', 0.27554897606711076), ('Criminal Traffic', 0.26463168981149027)]\
> \
> Recid(Cutoff=2y) Missing, Charge Severity Relative Frequencies\
> [(13.0, 0.17527002183155233), (18.0, 0.1732448580949098), (15.0, 0.14334137653682638), (14.0, 0.11454383545903711), (16.0, 0.07978570607836379), (17.0, 0.07804779960933012), (20.0, 0.06823796392048719), (11.0, 0.05019820751465012), (10.0, 0.0441514420314834), (12.0, 0.03172756520739975), (7.0, 0.015095369412846145), (21.0, 0.010800873262093532), (19.0, 0.009407675514190509), (9.0, 0.0060036768930253935), (8.0, 0.00014362863380443526)]\
> \
> Recid(Cutoff=180d) Missing, Charge Severity Relative Frequencies\
> [(13.0, 0.20459349729892964), (18.0, 0.11227896520484515), (15.0, 0.10870549944636446), (14.0, 0.09889105123645271), (10.0, 0.09825353152367211), (7.0, 0.0728953461060967), (12.0, 0.07141898466597323), (16.0, 0.06154581753514747), (17.0, 0.05608495789014529), (11.0, 0.05030533838875281), (20.0, 0.04114518672616851), (9.0, 0.011240479146394658), (21.0, 0.006400362379626212), (19.0, 0.006073214105962487), (8.0, 0.000167768345468577)]\
> \
> Recid(Cutoff=2y) Not Missing, Charge Severity Relative Frequencies\
> [(7.0, 0.3659044028356982), (10.0, 0.32531088725349827), (9.0, 0.10326693634742916), (13.0, 0.07375955968090224), (12.0, 0.051055783842318465), (15.0, 0.019611423796473213), (14.0, 0.01604868180678058), (11.0, 0.01307286141331573), (16.0, 0.011128772445665342), (18.0, 0.010385527906132336), (17.0, 0.008280852642177493), (20.0, 0.0010203624844831714), (19.0, 0.0006579774795483404), (8.0, 0.00042988809408935844), (21.0, 6.608197148811626e-05)]\
> \
> Recid(Cutoff=180d) Not Missing, Charge Severity Relative Frequencies\
> [(7.0, 0.3736417836796771), (10.0, 0.3308291996715166), (9.0, 0.10635939473616374), (13.0, 0.06747756406715497), (12.0, 0.04827675096022478), (15.0, 0.018133610261055935), (14.0, 0.013825763852830592), (11.0, 0.011707561378893836), (16.0, 0.01022275741941661), (18.0, 0.009790426107802954), (17.0, 0.007661175985358183), (20.0, 0.0009442056924850212), (19.0, 0.0006311889847579276), (8.0, 0.0004382233908179311), (21.0, 6.039381184381571e-05)]\

---

> > ### Author Response · Authors · 2023-08-25
> > **Response (Reviewer qVAQ)**
> >
> > 2. We discuss COMPAS/ProPublica dataset in Section 2. \
> > \
> > Further, Zafar et al 2017 report about COMPAS/ProPublica dataset: "The (unconstrained) logistic regression classifier leads to an accuracy of 0.668. However, the classifier yields false positive rates of 0.35 and 0.17, respectively, for blacks and whites (i.e., DFPR = 0.18), and false negative rates of 0.31 and 0.61 (i.e., DFNR = 0.30)."\
> > \
> > COMPAS/ProPublica dataset has the following number of samples from different racial groups. African-American:3175, Asian: 31, Caucasian:2103, Hispanic: 509, Native American:11, Other: 343. Due to very small numbers, racial groups other than African-American and Caucasian are generally not analysed in the literature using the COMPAS/ProPublica dataset. We hope this practice will also change with the availability of our dataset.\
> > \
> > Please see the significantly different results and number of samples for our dataset in Table 1 and Table 4. We also note in our experiments that the most disadvantaged group is not the 'African American' group but the 'American Indian or Alaskan Native' group (highest FPR and PR and lowest FNR).

---

> > > ### Author Response · Authors · 2023-08-25
> > > **Response (Reviewer qVAQ)**
> > >
> > > The above suggestions of the reviewer (descriptive analysis and comparison with COMPAS/ProPublica) will be added in the camera-ready version [comparison in the main paper, using the additional page NeurIPS allows upon acceptance and descriptive analysis in the supplementary material].

---

> > > > ### Author Response · Authors · 2023-08-25
> > > > **Response (Reviewer qVAQ)**
> > > >
> > > > 3. *"... the actual dataset file could benefit from an additional schema/key for the column names..."*\
> > > > \
> > > > We do have a file called 'description.md' in the dataset folder. Please let us know if the reviewer meant some other type of key.

---

### Official Review · Reviewer_byDS · 2023-07-11
**WCLD: Curated Large Dataset of Criminal Cases from Wisconsin Circuit Courts**

**Rating:** 8
**Confidence:** 4

**Strengths:**

The main strengths of the paper are:

1. the work provides a novel large-scale dataset that can greatly benefit the fair AI community
2. the dataset spans multiple years, allowing for studying data shifts
3. the limitations of the dataset are well-presented and are reasonable
4. some benchmarks are provided, allowing researchers to have a reference point for their evaluations


**Additional Feedback:**

I add here some comments/questions I have for the authors:
1. I think that adding some popular methods - such as LightGBM and Random Forest - to the benchmark could be (even more) beneficial to the community.
2. In section 4.2, can you explain how you preprocess categorical/numerical features for the classification task?
3. You state that  > 95% confidence intervals are constructed by multiple train and test splits. Can you please provide more details about this procedure?


**Clarity:**

The paper structure is clear and the dataset is well presented. I especially appreciated the explanations of why two target variables are provided.

**Correctness:**

The dataset creation is well-thought and all the choices seem in line with previous works. I have no complaints about it.

**Documentation:**

The dataset is properly curated and seems to address all required standards.

**Ethics:**

No issues.

**Limitations:**

The paper addresses its current limitations, such as the definition of the target variable.
These are quite common in the criminal justice literature and I personally think the authors took reasonable choices and properly documented them.


**Opportunities For Improvement:**

I really enjoyed reading the paper and think having access to such a source would greatly benefit the whole community.
The questions for the authors can be found in the Additional Feedback part.


**Relation To Prior Work:**

The dataset is well-framed to the fair AI literature.

**Summary And Contributions:**

The paper introduces a novel large dataset containing more than a million criminal cases from Wisconsin.
The dataset fills an important gap in the current literature on algorithmic fairness for criminal justice, which was mainly based on smaller-size datasets.

---

> ### Author Response · Authors · 2023-08-25
> **Response (Reviewer byDS)**
>
> We thank the reviewer for their time and valuable suggestions.
>
> 1. We are hoping to be able to put together an open source notebook later (separate from the paper), to help people use this dataset in their research. Other ML methods suggested by the reviewer will indeed be useful additions to such a notebook.\
> &nbsp;
> 2. We use one-hot encoding (using pandas get_dummies()) to process categorical variables for the classification task.\
> &nbsp;
> 3. The performance metrics are measured by training and testing the model multiple times with different random samples from the dataset, and confidence intervals are calculated from these measurements.

---

### Official Review · Reviewer_HBKu · 2023-07-25
**Court cases and recidivism dataset**

**Rating:** 5
**Confidence:** 5
**Correctness:** The authors don't make any miss-claim…
**Clarity:** The paper is cleary written.

**Strengths:**

Research ready, open-source, large criminal court datasets are rare and greatly needed, and this dataset will be a valuable addition.
I highly appriciate the efforts the authors have put into taking technically avilable raw court data and turn it into a research-ready dataset. I also appriciate the authors' intention to maintian the dataset and add valuable information such as bail details.

**Additional Feedback:**

None

**Documentation:**

I would have liked to see a longer section of potential uses and warning about miss-uses. Also, see the unclear points above.

The pre-processing and cleaning could be better described in the datasheet.

**Ethics:**

The ethics statement is lacking by missing to mention the debate about recidivism prediction and the limitations of algorithmic fairness (which is the stated intended use) within this space.

I am flagging for ethical review as I am concerned due to the use of recidivism as a target value, especially without proper treatment of relevant critical literature, errors and limitations and ethical risks.

**Limitations:**

The authors do a good job of listing some of the limitations of their dataset, such as the impact of not having jail/prison records. However, they do not cover in enough detail, in my opinion, some limitations of the data itself, such that convictions do not faithfully represent criminality and that offences committed outside of Wisconsin are not included, etc.

There are a few points that I was not clear on:

1. How, if at all, is mixed race being treated?

2. How is the past criminal record constructed? Is it from other cases in the raw data, or is it part of the data itself?

**Opportunities For Improvement:**

Without taking from the above, I have strong objections and concerns regarding the authors' choices of making general recidivism (i.e., not violent for example) the target variable for this dataset.

There has been much debate and well founded critism of the algorithmic fairness community using recidivism as a target variable without any criminal justice context (e.g., citation [11] in the paper), and this paper essentially encourages this practice.

Outside the research community, there have also been much criticm of the use of recdivisem predicators, even when they apprear to be unbiased (e.g., [1]), judges don't want of use them (e.g., [2]).

The authors themselves acknowledge that as they do not have prison / jail records, the reported recidivism is not accurate, and they are not  able to measure the extent of this inaccuracy. In addition, as they only have cases from Wisconsin Circuit Courts, they will also not capture recidivism elsewhere, also, an error that is hard to quanitfy the impact of.

You can see by the evaluation section that this dataset, despite (as the authors claim) having multiple improvements over the COMPAS dataset, does not improve the result of recidivism prediction models. This is because the limiting factors of such models are not due to insufficent data.

All of that said, while I appriciate that if published some people will use the dataset for recidivism prediction, I really think the authors should not define this as the target variable and focus thier paper on claiming thier dataset is an improvement to COMPAS. There are so many other useful use cases for such data. One can use a varity of methods (including causal ML, for example) to study the process of sentencing itself, or try and measure the impact of defendant attributes of the severness of sentencing or bail (if added). If you do want to focus on recidivism, please at least consider violent recidivism instead of general recidivism.

I also feel like the reader would benifit more from an evaluation section that is not focused on traning recidivism predictiors.

[1] https://dl.acm.org/doi/10.1145/3593013.3594099
[2] https://dl.acm.org/doi/10.1145/3593013.3593999

**Relation To Prior Work:**

There is a lot written about how to use criminal justice data responsibly within ML that would have been very useful but is excluded from this paper.

I would also expect a more thorough description of similar (court records) datasets.


**Summary And Contributions:**

The authors created a large dataset of court case decisions and metadata from criminal cases brought in front of Wisconsin Circuit Courts between 2000 and 2018. The authors further create a target variable `recidivism' based on re-appearance in from of a Wisconsin Circuit Court in the following 2 years.

---

> ### Author Response · Authors · 2023-08-25
> **Response (Reviewer HBKu)**
>
> We thank the reviewer for their time and thoughtful suggestions.
>
> 1. *Longer Section on Uses/Warnings, Citing Critical Literature and Ethics Flag*: As discussed in the intended use section as well as in the introduction, the dataset is for academic research. We have also listed several fundamental limitations of data (acknowledged by the reviewer) and algorithms in this context. The license is CC 4.0 BY-NC-SA (Attribution-NonCommercial-ShareAlike).\
> \
> To further address the concerns raised by the reviewer, we will remove the term "target" for describing the recidivism variable in the dataset. The sections 'Intended Use and Limitations' and 'Ethical Considerations' will be extended. More specifically, we will cite the related critical work suggested by the reviewer and discuss those. We will highlight other fundamental limitations in any recidivism variable construction suggested by the reviewer (in addition to the many we already mention). We will also mention other potential uses (beyond analysis of recidivism prediction task) of this data suggested by the reviewer.\
> \
> The changes will be implemented using the additional page allowed by NeurIPS for accepted papers.

---

> > ### Author Response · Authors · 2023-08-25
> > **Response (Reviewer HBKu)**
> >
> > 2. *"... can see by the evaluation section that this dataset, despite (as the authors claim) having multiple improvements over the COMPAS dataset, does not improve the result of recidivism prediction models. This is because the limiting factors of such models are not due to insufficent data ..."*\
> > \
> > We view this comment in a positive way: the fact that the reviewer is making this inference and supporting it with empirical evidence also shows the value of our dataset for conducting conclusive critical academic research.\
> > &nbsp;
> > 3. Question about mixed-race: As per WCCA declaration (included in the datasheet): "In criminal cases, any designation in any race field contains subjective information generally provided by the agency that filed the case."
> > We found that **for each case, the defendants are assigned exactly one race**. For a small number of defendants, **different cases** assign a different race to the same defendant. This doesn't necessarily imply mixed-race but could be (and perhaps more likely to be) a data quality issue. \
> > \
> > More specifically, for ~ 0.3% of the defendants, there were more than two races reported (across different cases). For another ~ 0.3% of the defendants, there were two races reported (across different cases) and none of the two races was caucasian. There was perhaps no obvious way to determine the race in these records. We did not include such records for fairness reporting and in the dataset. For ~5% of defendants, there were two races reported (across different cases) and one of the two races was caucasian. We observed that for most of these defendants, non-caucasian race is the most frequently reported race in cases associated with the respective defendant. For reporting the model (un)fairness, we used the non-caucasian race.\
> > \
> > We have now also created an additional column ('all_cases_races') that contains this additional race information for all the defendants in the dataset to ensure that users of the dataset will have access to this information.\
> > \
> > The dataset file has been updated to include the new column. The text in the paper pdf (including the datasheet) will be updated in the camera-ready version using the additional page allowed by NeurIPS for accepted papers.\
> > &nbsp;
> > 4. Question about past criminal count: It was not part of the original data and the process for constructing it is described in 3.3.1 (please also check the background sentences in 3.3 that explain how we performed database search).

---

> > > ### Author Response · Authors · 2023-08-25
> > > **Response (Reviewer HBKu)**
> > >
> > > 5. Suggestion regarding violent recidivism: As noted by the reviewer, it is not common to use violent recidivism information in the research community. Having said that, we agree with the reviewer's suggestion that including information on violent recidivism will be useful.\
> > > \
> > > We note that creating violent recidivism column also involves making certain choices because:\
> > > \
> > > i) In the FBI’s Uniform Crime Reporting (UCR) Program, violent crime is composed of four offenses: murder and nonnegligent manslaughter, forcible rape, robbery, and aggravated assault. Violent crimes are defined in the UCR Program as those offenses which involve force or threat of force. (https://ucr.fbi.gov/crime-in-the-u.s/2010/crime-in-the-u.s.-2010/violent-crime) \
> > > ii) To the best of our knowledge, in the information available from WCCA, there is no direct correspondence between charge class or wcisclass and violent/non-violent crimes.\
> > > iii) There are over 90000 different charge descriptions in felony and misdemeanor offenses in our dataset. Manually classifying crimes using such a large number of charge descriptions faces obvious challenges and limitations.\
> > > iv) To the best of our knowledge, there is no classification of violent crimes according to the statute cites in the Wisconsin Statutes.\
> > > \
> > > One alternative that we are exploring is the use of the following online tool for crime classification:\
> > > \
> > > https://cjars-toc.isr.umich.edu
> > > \
> > > [Choi, J., Kilmer, D., Mueller-Smith, M., & Taheri, S. (2023). Hierarchical Approaches to Text-based Offense Classification. Science Advances. 9(9), 1-15. ]\
> > > \
> > > However, we didn't find the source code for the tool online, therefore, we have to wait for the online tool to process the classification task, which is still in queue. We have also sent the authors a request for the source code but at the time of writing the rebuttal, we have not received a response. If the reviewer has any other suggestions, we will be happy to hear about them.

---

> > > > ### Author Response · Authors · 2023-08-25
> > > > **Response (Reviewer HBKu)**
> > > >
> > > > **Summary:** As noted by all the reviewers, the dataset is a valuable resource for the community in its current form too and a very significant improvement over alternatives (e.g. the COMPAS/ProPublica dataset). We have provided additional information asked by the reviewer that was feasible to obtain and added it in the dataset. We have committed to including additional ethics related text suggested by the reviewer, using the additional page provided by NeurIPS for accepted papers.\
> > > > \
> > > > We also welcome the suggestion of the reviewer to include a column on violent recidivism and we are working on it (described in detail in the previous comment). However, we believe that not having this additional column before the rebuttal deadline should not be not a reason for rejection of the dataset/paper.

---

> ### Comment · Ethics_Reviewer_UEuV · 2023-08-30
> **Ethics Review - addressing Ethics concerns raised by the technical reviewers**
>
> The reviewer's concern regarding an ethics statement for the feature of recidivism prediction is well-founded. There are multiple types of  well-documented bias found in machine learning models related to recidivism.
> The authors state that they are working on a further submission to address these issues.
> I would recommend a conditional-response to the authors contingent upon their stated further-submission addressing the reviewer's concerns.
>
> The NeurIPS Code of Ethics requires data and model documentation. https://nips.cc/public/EthicsGuidelines
> Although the determination of recidivism is known to be problematic, the authors are still required to address known standards and known comparative deficiencies in their dataset and model. Some of these issues are addressed in Section 3.5 of the Supplemental material, but it would be helpful if the authors' included information about the known bias limitations of recidivism overall and how their inclusion of the variable makes a contribution that limits the known bias in making a recidivism determination based on their expressly curated variable.
> Providing information about research artifacts contributing to the characterization of the metes-and-bounds of the recidivism variable in the dataset is critical to enable external-scrutiny and auditing.
>
> The data were publicly obtained, so there does not appear to be an issue of legal compliance or consent. Additionally, the public record of criminal history is sanctioned by the appropriate legal and governmental authorities in the state of Wisconsin. The dataset does not pose any concerns directly related to the overarching criminal database subject matter as it is in compliance with the law of the jurisdiction.
>
> As an additional note, the authors appear to make a substantial effort to stress that the dataset is for academic-research-use-only (see Section 5.3 of the Supplemental Materials). In Section C.6 of the supplemental materials, the authors state: "How will the dataset be distributed? The dataset will be hosted at the institute website of corresponding author Elliott Ash".
> However, there is no indication as to whether this dataset will be restricted to "academic-research-use-only" by a registration process or whether it will be freely available simply by accessing it on the stated website.
> Presumably there are no consent issues or other issues requiring restricted access to the dataset, so it is unclear why the limitation on academic-use-only if a public open-web link is going to be utilized, unless this recitation is to comply with GDPR. Clarification by the authors would be appreciated.
> If a registration process is required in order to provide evidence of academic affiliation or as a click-wrap agreement in the form of an academic-use-only acknowledgement, then the supplemental materials should be updated by the authors to reflect the same.

---

> ### Comment · Reviewer_HBKu · 2023-08-31
>
> I thank the authors for engaging with my review. I must say I am not sure what the authors mean by *critical* academic research, or how such a thing can be enforced. In addition, I don't understand how the concerns raised become less valid by restricting the use to academic research.
>
> I appreciate the authors' willingness to add violent recidivism and appreciate that this effort is beyond the rebuttal period. However, all the other changes mentioned, and the additional page of ethical discussions and considerations could have been reasonably done within this period, and I am unable to reconsider my score without reviewing these.

---

> > ### Author Response · Authors · 2023-08-31
> > **Revision Available**
> >
> > We thank the reviewer for reading our response. The reason we did not upload the revision is that we were not sure about NeurIPS policy on using the additional page during the review phase. However, it appears from your response (a few minutes ago) and the ethics review (which was only submitted a few hours ago) that the revision can be provided during the review phase. Accordingly, we have uploaded a revision on openreview. We request you to kindly take a look at the revision, all changes are marked in blue.
> >
> > The paper mentions intended use as academic research, which includes critical analysis. In our response, we did not mean that the concerns become less valid by restricting the use to academic research. This is why we committed to the revision, and have now also uploaded the revision.
> >
> > We look forward to your response and thank you once again for your time.

---

### Decision · Program_Chairs · 2023-09-22

**Decision:**

Accept (Poster)

**Comment:**

While there is the over looming concern of the use of recidivism as a target of research that spawns from this dataset, the contribution itself is worthy of publication in this venue due to its improvement over other similar datasets (COMPAS) and the additional, and beneficial evaluations that could lead to. Most of the responses to the other reviewers address most of the issues, but the clarification on how the dataset will be made available needs to be properly defined.